# Myocardial Cell Preservation from Potential Cardiotoxic Drugs: The Role of Nanotechnologies

**DOI:** 10.3390/pharmaceutics15010087

**Published:** 2022-12-27

**Authors:** Adelaide Iervolino, Luigi Spadafora, Cristiano Spadaccio, Valentina Iervolino, Giuseppe Biondi Zoccai, Felicita Andreotti

**Affiliations:** 1Department of Clinical Medicine and Surgery, Federico II University, 80131 Naples, Italy; 2Department of Clinical, Internal Medicine, Anesthesiology and Cardiovascular Sciences, Sapienza University of Rome, 00189 Rome, Italy; 3Department of Cardiovascular Surgery, Mayo Clinic, Rochester, MN 55905, USA; 4Centre Hospitalier Universitaire Henri-Mondor, Faculté de Médecine, Université Paris Est Créteil, 94000 Créteil, France; 5Department of Medical-Surgical Sciences and Biotechnologies, Sapienza University of Rome, 00189 Rome, Italy; 6Mediterranea Cardiocentro, 80122 Napoli, Italy; 7Department of Cardiovascular Sciences, Fondazione Policlinico Universitario A Gemelli IRCCS, 00168 Rome, Italy

**Keywords:** delivery systems, nanoparticles, liposomes, dendrimers, cardiotoxicity, cardio-oncology, cardiomyocytes

## Abstract

Cardiotoxic therapies, whether chemotherapeutic or antibiotic, represent a burden for patients who may need to interrupt life-saving treatment because of serious complications. Cardiotoxicity is a broad term, spanning from forms of heart failure induction, particularly left ventricular systolic dysfunction, to induction of arrhythmias. Nanotechnologies emerged decades ago. They offer the possibility to modify the profiles of potentially toxic drugs and to abolish off-target side effects thanks to more favorable pharmacokinetics and dynamics. This relatively modern science encompasses nanocarriers (e.g., liposomes, niosomes, and dendrimers) and other delivery systems applicable to real-life clinical settings. We here review selected applications of nanotechnology to the fields of pharmacology and cardio-oncology. Heart tissue-sparing co-administration of nanocarriers bound to chemotherapeutics (such as anthracyclines and platinum agents) are discussed based on recent studies. Nanotechnology applications supporting the administration of potentially cardiotoxic oncological target therapies, antibiotics (especially macrolides and fluoroquinolones), or neuroactive agents are also summarized. The future of nanotechnologies includes studies to improve therapeutic safety and to encompass a broader range of pharmacological agents. The field merits investments and research, as testified by its exponential growth.

## 1. Introduction: Nanotechnology Applications as Pharmacological Strategies

Cardiotoxicity has gained considerable attention in cardio-oncology. However, the term can be appropriately applied more broadly to any potential, drug-induced, cardiovascular side effect [1,2]. It represents an important cause of comorbidity and of therapy discontinuation at follow-up visits, especially for oncological patients. A current challenge is to offer the best and most up-to-date care to patients, while providing in parallel the best available tools to treating physicians [2]. Pharmacology is continuously evolving, with several active compounds marketed decades ago having been withdrawn and others repurposed; an emerging tendency is the co-administration of nanomolecules to improve certain drugs’ pharmacokinetic and pharmacodynamic profiles [3,4]. Nanoparticles meeting criteria of safety have been combined with drugs to potentiate their target interaction while reducing side effects and toxicities.

We here present nanotechnologies applied to the field of pharmacology, focusing on recent studies involving heart tissue-sparing co-administration of lipid-based nanocarriers with anthracyclines and platinum agents, macrolides, fluoroquinolones, and neuroactive agents. Several other types of nanoparticles have been tested for improved drug delivery and distribution, such as metal-based nanomaterials, carbon particles, magnetic field- driven particles, and nanogels [5,6,7]. Some of them under clinical investigation will also be discussed in the following sections.

### 1.1. Nanocarriers: A Subset of Nanostructures

Nanotechnology is a modern science contributing to planning, producing and distributing materials of nano-level dimension, notably between 10 and 1000 nanometers. Nanostructures can be applied to tissue engineering, regenerative medicine, imaging, theranostics (a combination word for therapeutics and diagnostics), and drug delivery/target therapy [8]. Cutting-edge delivery systems—nanocarriers—represent the most valuable application of nanotechnology. In particular, efforts are made to modify drug profiles and improve bioavailability in order to reduce dose requirements, improve distribution and reduce undesired side effects. Nanostructures have several attractive properties, such as a high surface-to-volume ratio [9]. An important function is the controlled delivery of drugs with respect to timing and specific targeting, so that the active pharmaceutical ingredient (API) is delivered exclusively to the desired site of action. Applications to anti-infective, oncological, and radiation therapies have been explored [10].

Nanocarriers are mostly represented by nanoparticles (including liposomes, niosomes, and dendrimers), nanoemulsions, and microemulsions. The latter two differ in the size of the particles dispersed in the liquid phase [11]. They can be synthesized to achieve safe oral drug delivery. Nanocarrier assembly can be obtained by polymerization reactions or by direct polymer synthesis [12]. Nanocarriers carry and protect the API against enzymatic and hydrolytic degradation and prolong holding time in the intestinal mucosa, thereby improving oral bioavailability [13]. In this review, we focus on liposomes and other nanocarriers shown to be associated with decreased cardiotoxicity and improved drug target delivery. While only properly structured clinical trials can unequivocally demonstrate reduced cardiotoxicity by nanotechnology formulations, we mostly report promising results from preclinical studies, as well as available clinical data.

### 1.2. Structure and Potential of Liposomal Carriers

Liposomes are amphiphilic (i.e., they contain both hydrophilic and hydrophobic groups), with a phospholipid bilayer membrane and a hydrophilic core. They can carry both lipophilic and hydrophilic drugs. The advantages of coating an API with a liposomal structure is to improve its therapeutic index (i.e., its safety or toxic-to effective-dose ratio) and efficacy through active and passive targeting effects [14,15]. Active targeting consists of coating nanoparticles with ligands that can interact, for instance, with receptors on tumor cells, thus enhancing their delivery to the tumor site [15]. Passive targeting is identified by enhanced permeability and retention (EPR) effects. It is the mechanism by which nanoparticles carrying high-molecular-weight drugs preferentially accumulate in inflamed tissues, given the high vascular permeability of the latter [7,8,9].

In most cases, liposomal carriers also improve drug bioavailability. Following ingestion, liposomes are resistant to intestinal enzymes and subsequently enter the circulation, reaching their pharmacological target site. A fundamental characteristic is their size: liposomes are often able to pass only through bigger pores in the endothelium, which usually reside at inflamed sites, especially tumoral ones, where the vasculature is greatly permeable [16]. Normal myocardial tissue, on the other hand, has very tight capillaries. Biodegradability of liposomal carriers is favorable, and the active compound is released rapidly. Liposomes are generally not capable of entering cells, so that the drug is released into the extracellular space. An ammonium sulphate gradient is used to counteract exceedingly rapid release of the drug [17]. Another strategy to prolong and accumulate drug dose is to cover the bilayer with neutral phospholipid polyethylene glycol (PEG) polymers—a process commonly known as PEGylation [18], which usually changes the structure and previous molecular bonds, enhancing solubility while reducing immunogenicity and degradation [19].

Liposomes have entered the pharmaceutical market and are approved for various target treatments in co-administration with anti-infective agents (e.g., ciprofloxacin, amikacin, and tobramycin) [20] and cancer therapies (e.g., doxorubicin for breast and ovarian cancer and vincristine for lymphomas and leukemia [21]). Liposomes are a key component of the mRNA-1273 vaccine against COVID-19 with clear reports of efficacy and safety [22].

### 1.3. Niosomes

Amphiphilic nanocarriers have been engineered with non-ionic surfactants and repurposed following adaptation of their composition, size, surface charge, and structure. These carriers, named niosomes, consist of a single-chain surfactant coupled with cholesterol to form vesicles. Although they possess less cholesterol than liposomes [23], niosomes may be easily modified in relation to their cholesterol fraction, the type of surfactant, and the “critical packing parameter”. The latter is a mathematical representation of surfactant aggregation determined by three parameters: lipid volume, optimal surface area, and critical chain length [24,25]. Important advantages of niosomes are the possibility to alter the distribution of the drug inside the vesicle, and lower cost and greater manageability compared to liposomes.

Niosomes have been tested as carriers of chemotherapeutic agents, anti-inflammatory drugs, and antibiotics. Rifampicin has been found to be effective against target bacteria when carried by niosomal preparations [26]. Studies in animal models have tested the potential advantages of niosomes combined with anti-inflammatory agents. Marianecci et al. [27] have loaded ammonium glycyrrhizinate, an organic molecule acting as a direct inhibitor of the high-mobility group box-1, into niosomes, showing good drug efficacy with low toxicity in experimental models and in human volunteers.

### 1.4. Dendrimers

Dendrimers are molecular structures with a very complicated architecture, designed to entrap APIs. Dendrimers have three sites of drug entrapment: void spaces, branching points, and surface molecular groups interacting and binding the drug intended to be carried [28]. They have an outer shell and an inner shell, can self-assemble and have low solubility [29]. They are quite versatile and can carry several molecules, with good bioavailability and a controlled-release delivery.

Dendrimer formulations have been approved as diagnostic tools and as therapeutic anti-infective options [30]. They are also under phase II clinical trial investigation in combination with docetaxel [31].

### 1.5. Carbon Nanomaterials

Carbon-based nanoparticles are inorganic materials, including nanodiamonds, nanohorns, nanodots, nanotubes, and graphenes, with desirable chemical properties such as inertness and biocompatibility coupled to unique electronic and thermal characteristics [32]. Carbon nanotubes are synthesized through various chemical methods: arc discharge, laser ablation, chemical vapor deposition, and ball milling [33]. These nanoparticles have been tested with paclitaxel and doxorubicin administration. Reduced doxorubicin toxicity, in terms of serum biomarkers and histopathological observations, has been reported after injection of a doxorubicin-conjugated carbon nanoparticle suspension in mice tumor models [34,35].

### 1.6. Nanogels and Hydrogel Polymers

Nanogels are crosslinked polymers of hydrogel materials at a nanoscale. Given their safety, amphiphilic behavior, and biodegradability [36], they are under investigation for both therapeutic and diagnostic applications. Nanogels can be classified according to their structures. The most important types are physically cross-linked, chemically cross-linked, and hybrid nanogels. Nanogels have been tested for preparations involving cisplatin for lung cancer [37], and hyaluronic acid nanogels targeting both CD44 and epidermal growth factor receptor (EGFR) for metastatic breast cancer [38].

### 1.7. Metal-Based Nanomaterials and Other Inorganic Particles

The main advantages of metal nanoparticles are good bioavailability, high drug payload, and surface chemistry [39]. The most used metals are gold, nickel, platinum, iron, and titanium dioxide, as they confer high solubility to the hydrophobic active ingredient [40]. Further surface modifications can be achieved by PEGylation.

Metallic nanoparticles have been used for chemotherapeutics and antibodies. Some of them have been approved by the Food and Drug Administration for prostate cancer (magnetic thermoablation) and for recurrent glioblastoma (metal nanoparticles covalently bound to spherical nucleic acids targeting the BCl2 gene) [40,41,42]. Despite promising ongoing clinical trials, a potential cardioprotective effect has still not been demonstrated for metal nanoparticles. 

Other types of nanoparticles, such as quantum dots or polystirene and magnetic and ceramic particles, have been recognized as optimal miniature-sized particles [39].

### 1.8. Magnetic Nanoparticles: Exploiting Their Physical Properties

Magnetic nanoparticles are particles of a nanosize dimension designed to move through imposed external magnetic fields, given their ferromagnetic, ferrimagnetic, and superparamagnetic properties [43,44]. Individual particles can be clustered together to form larger nanobeads [45]. Each particle typically presents an internal core with a quantum effect, formed by elements such as Fe oxides, Ni or Co, and a stabilizing coating to enhance surface efficiency [43]. The latter also contributes to the controlled release of the drug. These particles can be coated by several molecules such as glucan, chitosan or liposomes polyethylenimine and can also be pegylated [43].

Magnetic nanoparticles exhibit diverse applications in the theranostic field, such as drug delivery, radiotherapy, and magnetic resonance imaging [46,47].

Recently, Fe_3_O_4_, CoFe_2_O_4_, and ZnFe_2_O_4_ nanoparticles have undergone intense investigation. In particular, Fe_3_O_4_ has been used for liposomal vesicles, hydrogels or exosomes in cancer therapy and mesenchymal stem cell therapy engineering [48,49]. Their technical advantages include low toxicities, extraction without particle loss, and high retention time in the target site [50,51,52].

## 2. Myocardial Toxicity Induced by Oncological Drugs and Prevention by Nanotechnologies

Cardiotoxic oncological therapies include chemotherapeutic agents (such as anthracyclines), antiangiogenic factors, and other target therapies. The cardiotoxic effects are exerted because target molecules are also present on myocardial and endothelial cells [53]. As stated, nanomolecules in use or in development help reduce toxic side effects by passive targeting (liposome-enveloped drugs are developed into larger dimensions, so they can enter only more permeable capillaries, typical of tumor environments [54]) and/or active targeting (liposome ligands can link tumor cell-specific antigens) (Figure 1). For example, in mice, liposomal carriers coated with hyaluronic acid and loaded with the chemotherapeutic agent paclitaxel have been investigated for drug delivery to cancer cells with overexpression of CD44 [55]. Other preclinical studies have found effective delivery of nanocarriers to prostate tumor tissues [56]. We review relevant cardiotoxic chemotherapeutic agents according to the cardiotoxic mechanism of action and for each describe the nanotechnology applications shown to reduce cardiac side effects. Of note, most nanosized particles are under preclinical investigation. A few have successfully passed clinical trial testing, gained international drug agency approval and are finally marketed.

### 2.1. Anthracyclines

Anthracyclines exert cytostatic and cytotoxic actions against tumor cells by inhibiting type II topoisomerase and by subsequently inhibiting double-stranded DNA cuts and re-ligation [57]. Anthracyclines may also induce free radical formation and mitochondrial damage [58]. Well-known agents are doxorubicin, daunorubicin, epirubicin, and idarubicin. Cardiotoxic effects, ranging from mild to severe, include myocarditis, pericarditis, and heart failure. Such effects were believed to be caused by free oxygen radical formation, but this hypothesis has been dispelled [59]. Cardiomyocyte damage is caused by type II topoisomerase inhibition, increasing double-stranded human DNA breaks [60]. Interestingly, anthracyclines target both topoisomerase 2A and 2B forms, with the former being expressed by tumor cells and the latter being present constitutionally in all cells, including cardiomyocytes. By binding to this enzyme, anthracyclines can stop gene transcription regulating mitochondrial biogenesis, leading to mitochondrial dysfunction and ultimately to cell death [60].

Nanotechnology formulations have revolutionized anthracycline administration. Various strategies have been developed, including PEGylated and non-PEGylated liposomes, polymeric micelles, curcumin co-administration, dendrimers, and chitosan nanoparticles, all aiming at increasing specific delivery of the active compound by targeting the tumor microenvironmental niche, while maintaining good bioavailability and favorable kinetics. Typically, plasma drug concentrations do not reach high peaks but increase gradually to achieve delayed maximum concentrations [61,62,63].

Most systems have been investigated in preclinical in vitro or animal models. PEGylated liposomal doxorubicin (PLD) has undergone clinical trials and is approved for clinical use, representing a reference point of care for epithelial ovarian cancer, metastatic breast cancer, multiple myeloma, and specific forms of Kaposi’s sarcoma [64,65]. Regarding other nanoparticle formulations, chitosan has been investigated in mice tumor models. Mansour et al. [66] found improved lactic dehydrogenase and CK-MB values in hepatocellular carcinoma-bearing mice treated with doxorubicin-containing chitosan nanoparticles and verapamil, compared to values in mice treated with standard unloaded doxorubicin and verapamil.

Doxorubicin delivery by polymeric micelles (amphiphilic nanostructures with a hydrophilic core and a hydrophobic surface disposed around the active compound to be carried) [67] has also been investigated. Khaliq et al. [68] investigated intratumoral administration of the prodrug (DEVD-S_DPX), i.e., inactive doxorubicin linked to a peptidic part. Poloxamer nanoparticles (amphiphilic copolymers with the molecular structure polyoxyethylene–polyoxypropylene–polyoxyethylene (PEO–PPO–PEO) [69] containing a doxorubicin prodrug can transition from a solid state to a gel form, increasing retention in the tumor environment. Cote et al. [70] tested in vitro poloxamer micelles loaded with doxorubicin hydrochloride, resveratrol, and quercetin, with the latter representing potentially cardioprotective free radical scavengers. In healthy mice, administration of the chemical combination versus doxorubicin alone was cardioprotective.

Other promising nanostructures for anthracycline delivery in animal models are dendrimers. For most of these drug combinations, however, further clinical studies are necessary to prove their incremental efficacy and safety in man.

### 2.2. Cisplatin and Platinum Agents

Platinum agents, used to treat several cancer phenotypes (e.g., non-small cell lung cancer, head and neck cancer, and ovarian cancer), are a cornerstone of treatment for several reasons, including strong tumor-cell toxicity.

Cisplatin ((SP-4-2)-diamminedichloridoplatinum(II)) exerts its cytotoxic action by forming DNA adducts on purine bases, producing in turn cross-linking effects. Thus, cells are blocked in phases S, G1, or G2-M. Limitations to clinical use include poor drug retention inside cancer cells, drug resistance, and toxic side effects [71]. The latter include nephrotoxicity by cisplatin, myelosuppression by carboplatin, cytopenias and hepatotoxicity by both [72]. Other dose-limiting side effects include oxidative stress and myocardial mitochondrial damage, with possible development of irreversible cardiomyopathy. Watchers et al. [73] demonstrated higher rates of left ventricular ejection fraction deterioration, assessed by echocardiography, in patients already diagnosed with cardiac structural disease taking cisplatin rather than epirubicin combined with gemcitabine. Nanoliposomal formulations of cisplatin (termed lipoplatin) are part of standard oncological care, having passed the clinical trial phase, showing high dose accumulation in tumor sites, where they are thought to act as both chemotherapeutic and antiangiogenic agents [74]. Other strategies include gold nanocarriers and physical force methods such as magnetism, ultrasound, and heat [75,76]. Several studies were conducted for the development of cisplatin-loaded chitosan nanoparticles with successful results in terms of in vitro cytotoxicity and kinetics [77,78,79]. Animal studies also investigated the role of curcumin nanoparticles in reducing cardiotoxicity. Khadrawy and colleagues [80] showed decreased toxic effects, in terms of lipid peroxidation, nitric oxide, and tumor necrosis factor-alpha, in cardiomyocytes of rats treated with curcumin nanoparticles and a cisplatin intraperitoneal injection, compared to those in control rats receiving either saline or cisplatin only [80].

### 2.3. Cyclophosphamide

Cyclophosphamide is an alkylating agent that crosslinks with DNA at the guanine N-7 position, leading to cell apoptosis. The active metabolite is 4-hydroxycyclophosphamide, which exists in equilibrium with aldophosphamide and is oxidized, protein-bound, and transported to all tissues, including the placenta. Cardiotoxicity is by direct injury to capillaries, oxidative stress, endothelial and myocardial damage, toxic metabolites, edema, and microthrombosis [81]. Effects include reduced cardiac function, reduced ECG voltage, and pericardial effusion, which should be considered during follow-up of treated patients.

Suggestions of safer cyclophosphamide delivery to tumor sites are derived from in vitro studies of pH-sensitive carbonate apatite nanoparticles loaded with cyclophosphamide [82]. PEGylated niosomal formulations of cyclophosphamide (Nio-Cyclo-PEG) have also been described in in vitro studies, successfully targeting gastric cancer cells, with controlled release of the drug, prevention of cancer cell migration, and induction of tumor cell apoptosis [83].

### 2.4. Target Therapies: Bevacizumab and Trastuzumab

Bevacizumab and trastuzumab are among the best known target therapies causing possible cardiovascular side effects. Bevacizumab is a recombinant humanized monoclonal antibody against vascular endothelial growth factor, which is involved in angiogenesis. It was initially approved for metastatic colon cancer and subsequently for metastatic breast cancer, non-small cell lung cancer, glioblastoma, and ovarian cancer [84]. A common cardiovascular side effect is hypertension [85]. Some investigators have embedded bevacizumab into PEGylated cationic liposomes [86] and others into nano- and microspheres of polylactide-co-glycolic acid (PLGA) and polyethilene glycol-b-poly-lactic acid (PEG-b-PLA) acid [87]. Other nano-drug delivery systems include albumin nanoparticles and polysaccharide-based nanoparticles.

Trastuzumab is a monoclonal antibody directed against human epidermal growth factor receptor 2 (HER2) used to treat specific breast cancer cases and metastatic gastric cancer [88]. The therapeutic mechanism of action relies on drug binding to the extracellular domain IV of HER2, which triggers an intracellular cascade of tumor-suppressive actions including inhibition of oncogenic cellular signaling, and downregulation of angiogenesis and DNA repair pathways [89]. The hypothesized cardiotoxic trigger in humans is the accumulation of toxic reactive oxygen species in cardiomyocytes [90]. The clinical counterpart is impaired left ventricular function and arrhythmogenesis caused by altered membrane ion assets.

Investigated delivery particles include trastuzumab-loaded PLGA nanoparticles tested in vitro on HER2 positive breast cancer cell lines (SKBR3), with successful tumor suppressive results [91]. Dziawer et al. engineered gold nanoparticles labeled with α-emitting astate (^211^At) conjugated to both PEG and trastuzumab [92]. After injection in tumoral sites, the 5-nm-diameter preparation revealed, at microscopic levels, successful release and retention of ^211^At-AuNP-PEG-trastuzumab inside tumor cell, in proximity of the nucleus [92]. Again, these results must be interpreted as promising preclinical studies.

A selection of culprit cardiotoxic agents and their potential nanotechnology applications are presented in Table 1.

## 3. Myocardial Toxicity Induced by Antibiotics and Possible Remedies

### 3.1. Macrolides and Fluoroquinolones: Specific Applications

Antibiotics are globally recognized as fundamental pharmaceutical agents but with potential toxic side effects. Firstly, they affect the cardiac conduction system with possible arrhythmogenic effects. Macrolides and fluoroquinolones can induce tachyarrhythmias secondary to QT interval prolongation in both healthy subjects and patients [93]. These drugs carry a risk for torsade de pointes, ventricular tachycardia, and sudden cardiac death. The reasons are attributed to reactive oxygen species formation, mitochondrial membrane permeabilization, and mitochondrial swelling, as documented in rat cardiomyocytes [94].

Macrolides are bacteriostatic inhibitors of protein synthesis. Their indications include staphylococcal, Mycoplasma, Legionella, and Chlamydia infections [93]. Well-known side effects are gastrointestinal and hepatic, in addition to arrhythmogenesis. Since the first formulation of erythromycin, second- and third-generation compounds have shown improved pharmacokinetics leading to diminished resistance. However, new formulations, such as telithromycin, may induce dangerous liver injury [95]. Nanotechnologies have been shown to play a role in reducing certain side effects. It has been demonstrated that azithromycin in liposomal formulations favors cardioprotection. Liposomal azithromycin has been compared to free azithromycin in a post-myocardial infarction mouse model, showing reduced cardiac neutrophil and monocyte infiltration, as well as improved survival [96].

The commonly used fluoroquinolones are safe drugs exhibiting rare side effects. On a molecular basis, they are implicated in collagen degradation predisposing to aneurysms, aortic dissection, and tendon damage, in addition to arrhythmogenesis through QT prolongation. The latter is related to closure of the voltage-gated potassium channel IKr (the rapid component of the delayed rectifier potassium current) [97]. These sequelae are directly associated to administered dose and to serum concentrations [97]. Grepafloxacin and sparfloxacin were marketed and withdrawn in the 1990s because of sudden cardiac death events, as reported by population studies [98]. Levofloxacin has been associated with QT prolongation-related ventricular arrhythmias or cardiac arrest in extremely rare cases. Reports of torsade de pointes have not fully established a direct relation to levofloxacin [99].

Polymeric nanoscale approaches have been successfully applied to levofloxacin administration [100]. Further nano-approaches to reducing side effects have included chitosan, PLGA, albumin, arginine, and other organic and inorganic nanostructures. Chitosan, a chitin-derived polysaccharide, has been used for encapsuling drugs such as ciprofloxacin. These nano-microparticles were demonstrated to effectively deliver drug to lungs to target Escherichia coli, Pseudomonas aeruginosa, and Staphylococcus aureus [101]. Specific targeting by chitosan−dextran sulphate nanocapsules loaded with ciprofloxacin has also been reported against Salmonella-containing vacuoles [102]. The authors postulate the efficacy of this approach for other vacuolar microbial agents such as Mycobacteria, Brucella, and Legionella. Liposomes and micelles can enhance lung tissue permanence. Preclinical studies have demonstrated the efficacy of these delivery systems, and a phase II clinical trial reported low toxicity profiles coupled with potent antimicrobial effects and slow release [103].

Cotrimoxazole predisposes to hyperkalemia in elderly patients, especially during angiotensin-converting enzyme (ACE) inhibition and angiotensin receptor 1 blocker co-administration [104]. Large population studies have determined that sudden death was increased in patients taking cotrimoxazole mostly for urinary tract infections, and frank hyperkalemia was noted in a consistent subpopulation; the authors postulated that dangerous hyperkalemia-induced arrhythmias were responsible for most deaths. Liposomes have been investigated in relation to several other antimicrobial agents, including amikacin, aztreonam, and colistin [105].

### 3.2. Inhaled Liposomal Antibiotics: Efficacy and Drawbacks

Liposomal formulations of antibiotics can be delivered through direct nebulization to treat lung infections. Usually, pressurized metered-dose inhalers, dry powder inhalers, and medical nebulizers are the most used systems [106,107]. Liposomal amikacin, tobramycin, ciprofloxacin, and amphotericin B have been tested through inhalation (the latter to prevent pulmonary fungal infections). Bassetti et al. suggest these formulations for difficult-to-treat acute and chronic respiratory infections [108]. They allow higher concentrations in lung tissue and limit the toxic effects [109,110]. Different formulations reflect different pH, viscosity, surface tension, and osmolality. Amphotericin B rarely causes cardiac involvement, but when it does, critical cardiomyopathies and arrhythmias may develop [111,112]. The above new drug preparations may help reduce toxic side effects, especially in bedridden patients with frequent cardiovascular comorbidities.

## 4. Cardiovascular Toxicity of Drugs Acting on the Central Nervous System

Given similar electrophysiological processes occurring in both the heart and central nervous system (CNS), including action potential propagation and ion channel activation, drugs affecting the CNS may also affect cardiovascular tissues by targeting common biological pathways [113]. This biological commonality, however, is not the only aspect involved in the cardiotoxicity of CNS-acting agents, as they may also cause direct toxic damage, leading to myocarditis and heart failure [113,114]. Moreover, cardiovascular diseases (CVD) and psychiatric/neurological diseases may coexist: psychiatric disorders, for instance, can be both a risk factor or a consequence of CVD. Drug−drug interactions between cardiovascular- and nervous system-acting agents are not rare, as between oral anticoagulants and antiepileptic drugs or between benzodiazepines and statins [114]. Given the prevalence of these clinical conditions, it is important to evaluate the molecular mechanisms and clinical effects of cardiovascular toxicity related to the main CNS-acting agents (Table 2).

### 4.1. Antipsychotic Agents

Antipsychotic drugs are used most commonly against schizophrenia [115]. Second-generation agents have fewer neurological drawbacks than first-generations drugs [114,115]. Cardiovascular side effects include tachycardia (due to reduced vagal tone), bradycardia, and orthostatic hypotension (especially in elderly people), but the most feared effect is QT interval prolongation [115]. This side effect is due to alteration of potassium currents causing delayed repolarization, which may in turn lead to torsade de pointes, other arrhythmias, and sudden cardiac death [113,115]. Calcium and sodium currents may also be affected. The agents with most impact on the QT interval are haloperidol, pimozide, sertindole, and ziprasidone [113,114,115]. Direct cardiac damage by antipsychotic agents has been described for clozapine, associated with myocarditis of undefined underlying mechanism; other antipsychotic drugs may cause dilated cardiomyopathy, ventricular hypertrophy, and pulmonary thromboembolism [113,114,115]. The molecular mechanisms of cardiovascular damage by antipsychotic agents are manifold, including ion channel blockade, mitochondrial damage, and lack of lysosomal drug protonation [114]. Nanotechnologies have not been applied to antipsychotic drugs to provide cardioprotection. Third-generation antipsychotic agents emerging on the market, however, are reported to have fewer cardiovascular side effects than older ones [115].

### 4.2. Antidepressant Drugs

Antidepressant medications are used to treat not only depression, but also anxiety, obsessive compulsive disorders, and, in some cases, chronic pain [113,116]. Older drugs, such as tricyclic and neuroleptic agents, are known for being arrhythmogenic due to sodium, calcium, and potassium channel blockade [113]. Newer antidepressants, including selective serotonin reuptake inhibitors (SSRIs) as well as serotonin antagonists and reuptake inhibitors (SARIs), have fewer cardiovascular effects than older ones [117]. Trazodone, in the SARI class, has been investigated in cell cultures, showing sodium channel and human ether-a-go-go-related gene (hERG) inhibition as the basis of QT prolongation and related ventricular tachycardia [117]. The hERG codes for the alpha subunit of a potassium channel. SSRIs, even if with lesser intensity compared to first-generation antidepressants, inhibit sodium, calcium, and potassium channels, with possible associated QT prolongation, atrial fibrillation, or bradycardia [118]. Serotonin reuptake blockade may itself impair heart conduction, causing ventricular arrhythmias or bradycardia [116]. Other specific molecular pathways by which SSRIs interact with cardiac cells include the L-type calcium current, IKr and hERG trafficking for fluoxetine, and inward sodium current blockade for venlafaxine [113]. As for antipsychotic drugs, no specific cardioprotection has been described for these medications, although newer agents target the CNS in a more specific way than older ones.

### 4.3. Local Anesthetics

Sodium channels, present in both cardiac and central nervous tissue, are targeted by local anesthetics [113]. The cardiovascular effects of cocaine, the first local anesthetic, have been extensively studied and include impaired calcium current, reduced nitric oxide bioavailability, and increased myocardial oxygen demand; these effects may lead to myocardial infarction, a prothrombotic state, aortic dissection, and stroke [119]. Newer-generation local anesthetics, bupivacaine and lidocaine, developed to overcome the above side effects, do not prove exempt from severe cardiovascular effects (hypoxia and arrhythmias) [113]. Mitochondrial membranes, through enhanced cardiolipin permeability, seem to play a role in local anesthetic cardiotoxicity [113]. To resolve the problem of resistant cardiotoxicity induced by local anesthetics, the so-called “lipidic reanimation” strategy has been investigated, consisting of intravenous lipid emulsion administration, with encouraging results [120].

### 4.4. Other Agents

Dopamine receptor agonists, administered in Parkinson’s disease, can cause orthostatic hypotension, peripheral edema, and arrhythmias [113]. Among these agents, pergolide has been linked to myocardial fibrosis and fibrotic valvulopathy [113]. Pergolide-mediated fibrosis involves serotonin 5-hydroxytryptamine receptor 2B and transforming growth factor-beta [113,121]. Other CNS-acting agents that can cause myocardial fibrosis include methysergide, ergotamine, and dexfenfluramine [113]. Opioid analgesics, in particular levomethadyl which was removed from the market, may impact cardiac electrophysiology and hemodynamics, due to hypotensive actions [113].

### 4.5. How to Overcome Cardiotoxicity Induced by CNS-Acting Agents

Avoiding cardiotoxicity induced by CNS-acting agents is challenging, as it is estimated that one in six people suffers from CNS disorders requiring specific drugs [121]. Despite the size of the problem, to date, few efforts have been focused on the application of nanotechnology to prevent cardiotoxicity related to CNS-acting agents [113]. The blood−brain barrier (BBB), made of endothelial cells, basement membranes, and glial cells with their projections, prevents the passage of large molecules and many small molecules, ensuring strong protection of the CNS [122,123]. This protection, however, is the basis for suboptimal selectivity of CNS-acting agents [123]. The filtering action of the BBB is performed by three main mechanisms: carrier-mediated transporters, active efflux transporters, and receptor-mediated transporters [124]. To promote selective delivery of drugs to the CNS, peptide shuttles have been developed [124]. These, combined with drugs thanks to linker molecules, are able to target specific epitopes of the CNS, ensuring neuroselectivity while reducing unwanted effects in other districts [121]. In addition to shuttle peptides, single-domain antibodies derived from the heavy chains of antibodies of camelidae and cartilaginous fish have also proved capable of crossing the BBB and might be exploited further [125]. These potential solutions to ensure neuroselectivity are additional to the mechanisms described above for anticancer drugs. Research objectives in this area include the following: further implementation of available technologies (peptide shuttles, single-domain antibodies, liposomes, etc.); improved understanding of the transport mechanisms across the BBB; and identification of new molecular targets (epitopes and membrane proteins) specific for the CNS.

## 5. From Reduced Cardiotoxicity to Cardioprotection: Getting Closer to Precision Medicine

### 5.1. Repurposing Cardiovascular Drugs

The use of cardioprotective drugs during cardiotoxic therapies, both oncological and non-oncological, has gained increasing interest. Several cardiovascular event-preventing drugs have been considered [126], including ACE inhibitors, angiotensin receptor blockers, and beta-blockers. In particular, carvedilol, a non-selective beta and alpha-1 adrenergic receptor blocker with several indications (e.g., hypertension, arrhythmias, and heart failure) [127], demonstrated effective prevention of doxorubicin-induced cardiovascular complications, as assessed by echocardiographic strain parameters [128]. Sacubitril/valsartan, a combination therapy of angiotensin receptor/neprilysin inhibitors improving the prognosis of patients suffering from heart failure with reduced ejection fraction [129,130], is currently under investigation as a preventive agent during cardiotoxic therapies, although data from large clinical trials are missing.

Dexaroxane hydrochloride, also called cardioxane, is an iron chelator capable of mitigating free oxygen radical formation by binding free iron and removing it from doxorubicin−iron complexes [131]. It has recently been approved for children and adolescents undergoing high therapeutic doses of doxorubicin [132]. Statin use has also been investigated in the preventive setting during treatment with anthracyclines, whereas glifozines, or SGLT-2 (sodium/glucose cotransporter-2) inhibitors, approved in 2020 for the treatment of heart failure even without concomitant diabetes, are under investigation for cardioprotection of frail patients during oncological and non-oncological therapies [133,134,135].

### 5.2. Limitations

An important current challenge of nanotechnologies is the safety and reproducibility of laboratory techniques when transferred to humans. Sterility achievement may be an issue, given that high temperatures can disrupt the chemical stability and thus the nanoparticle comformation [136]. Properties such as stability, biodegradability, absorption, and solubility may vary at variable temperatures and pH [137]. Drug accumulation may occur in case of prolonged treatments. In some cases, cytotoxicity has been observed after nanoparticle administration. Metal-oxide nanoparticles can generate oxygen free radicals, inducing mitochondrial damage, as shown for cells exposed to carbon nanoparticles [138]. Nanoparticles (particularly metal, such as gold, zinc oxide, and iron oxide) may alter intracellular signal cascades and cell membrane stability, such as calcium homeostasis, as reported in in vitro lung epithelial cancer cells exposed to zinc nanoparticles [139]. Carbon-based nanoparticles may cause serious side effects including carcinogenesis, nephro and cardiotoxicity, impaired immunogenesis, neurotoxicity, hepatotoxicity, and bone toxicity through mitochondrial and DNA damage [140,141]. Polymeric nanocarriers have been reported to be relatively safe, non-inflammatory, and non immunotoxic [142,143].

Other issues regard preclinical study designs related to the administered dose of active drug in animal tumor models. While human tumors usually present a low tumor/body weight ratio, animal tumors are usually large with respect to the overall size and thus the EPR effect can be overestimated [144,145]. Moreover, higher doses are generally administered to animals compared to those administered to human patients, given that the maximum tolerated dose is higher in animals [146]. Thus, translation of animal studies to the clinical setting may be difficult.

### 5.3. Future Perspectives

Nanocarriers and nanomolecules may also be used to deliver protective agents, such as resveratrol, during cardiotoxic treatments. Resveratrol consists of a polyphenolic phytoalexin exhibiting cardioprotective effects against postmyocardial ischemic reperfusion injury and ventricular arrhythmias; however, it shows poor solubility, photosensitivity, and rapid metabolism, with suboptimal bioavailability [147,148]. It has gained attention for its ability to sensitize cancer cells to chemotherapeutic agents [149]. Animal studies additionally report antidiabetic and cardiotonic effects of resveratrol, coupled with good safety profiles, which have opened the way for clinical trials [150] Several nano-based formulations have been investigated including polymeric nanoparticles, liposomes, cyclodextrins, and micelles [151].

Cardioprotective micro-RNAs (miRNAs) are being investigated in preclinical stem cell studies [152]. Cardioprotective miRNA studies have employed a remote ischemic preconditioning model, exposing myocardial tissue to brief non-fatal ischemia and observing improved LVEF, as well as reduced apoptosis, arrhythmias, and fibrosis, by some miRNAs [153,154,155,156]. The most important investigated miRNAs are miR-665, miR-132, miR-126-5p, and miR-210 [157]. MiRNAs are a crucial component of biochemical inflammatory pathways and merit further preclinical investigations to understand their therapeutical potential.

Natural compounds, such as curcumin, quercetin, and hesperetin, have also gained recent attention for their beneficial properties and have been incorporated into nanocarriers to favor their kinetic profiles [158,159,160]. Curcumin, in particular, has shown anti-hypercholesterolemic, anti-atherosclerotic, and protective effects against cardiac ischemia and reperfusion damage. The molecule has been tested in clinical models despite its low bioavailability [161]. Nanotechnologies to enhance its bioavailability have combined natural phytochemicals with liposomes, dendrimers, polymers, and other nanoparticles, targeted to myocardial and endothelial cells.

## 6. Conclusions

The use of nanotechnologies is an expanding and challenging field, especially for the delivery of drugs that need specific targeting and have potentially toxic effects in off-target tissues. Some formulations have been approved and are used in clinical practice, while in many cases ongoing and published studies are still at a preclinical level or limited to in vitro experiments. Clinical studies may be limited by investigations being performed in patients undergoing cardiotoxic treatments with an already established myocardial injury, providing researchers with population selection biases and statistical concerns.

Nonetheless, comprehensive clinical data and metanalyses-based evaluations should be encouraged to reach a higher level of knowledge and enable physicians to elaborate consensus documents about the safety and on-label use of nanotechnologies. Advantages of nanotechnologies include the relatively low cost of the synthesis processes and the reduced sized products. Nanoproducts can potentially offer high stability, specific targeting, accuracy of action, maintained efficacy, and improved safety of drugs. The controlled release of the carried medication is a further opportunity to limit potential side effects [162,163,164].

Heart-sparing nanodelivery systems under evaluation in the field of cardio-oncology are based on relatively safe materials such as lipids or polymers. However, caution is needed, as successful results in preclinical studies do not always translate into therapeutic success. To further characterize the impact and potential advantages of nanoparticles, further research and investments should be dedicated to this fascinating new specialty.

## Figures and Tables

**Figure 1 pharmaceutics-15-00087-f001:**
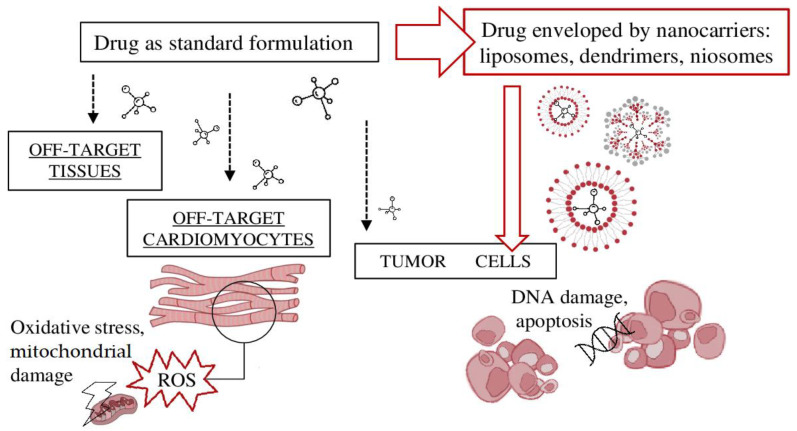
Nanotechnology drug strategies in oncology. (**Left**) standard active compounds acting on both the tumor target site and off-target cells, causing DNA and mitochondrial damage and cell death; (**right**) drug-carrying delivery systems acting only on tumor cells, given specific conformation and tumor capillary fenestrations. Abbreviations: DNA, deoxyribonucleic acid; ROS, reactive oxygen species.

**Table 1 pharmaceutics-15-00087-t001:** Cardiotoxicity of selected chemotherapeutic agents and target therapies: molecular mechanisms and potential nanotechnology applications. Most investigations are currently in a preclinical stage. Abbreviations: DNA, deoxyribonucleic acid; G2-M, Gap2-Mitotic phase checkpoint; HER2, human epidermal growth factor receptor 2; PEG: polyethylene glycol; PLGA, polylactide-co-glycolic acid.

Drug	Cardiotoxic Molecular Mechanism	Cardiovascular Side Effects	Nanotechnology Applications
Anthracyclines	Inhibition of type II topoisomerase	Myocarditis, pericarditis, and heart failure	PEGylated and non-PEGylated liposomes, polymeric micelles, dendrimers, chitosan, or poloxamer nanoparticles
Cisplatin and platinum agents	DNA adducts on purine bases; Cells blocked in phases S, G1, or G2-M; Myocardial mitochondrial damage	Cardiomyopathy, left ventricular ejection fraction deterioration	Liposomes, chitosan nanocarriers, and metallic or curcumin nanoparticles
Cyclophosphamide	Direct cell injury to capillaries, oxidative stress, endothelial and myocardial damage, edema, and microthrombosis	Reduced cardiac performance, reduced ECG voltage, and pericardial effusion	Carbonate apatite nanoparticles and PEGylated niosomes
Bevacizumab	Antagonism of the vascular endothelial growth factor	Hypertension	PEGylated liposomes, albumin nanoparticles, and PLGA microspheres
Trastuzumab	Antagonism of HER2	Accumulation of toxic reactive oxygen species in cardiomyocytes; reduction of left ventricular ejection fraction	PLGA or gold nanoparticles

**Table 2 pharmaceutics-15-00087-t002:** Cardiotoxicity of agents acting on the central nervous system: molecular mechanisms and manifestations. Abbreviations: NO, nitric oxide; TGF-b, transforming growth factor-beta.

Drug	Cardiotoxic Molecular Mechanism	Cardiovascular Effects
Local anesthetic agents	Sodium channel blockade, reduced NO bioavailability, increased mitochondrial permeability to cardiolipin, and increased myocardial oxygen demand	Arrhythmias, myocardial infarction, stroke, and aortic dissection
Anti-depressant agents	Sodium, calcium, and potassium channel blockade and Herg trafficking blockade	Ventricular arrhythmias, bradycardia, and QT interval prolongation
Anti-psychotic agents	Ion channel blockade, mitochondrial damage, and lack of lysosomal drug protonation	Tachycardia, bradycardia, orthostatic hypotension, QT interval prolongation, myocarditis, dilated cardiomyopathy, ventricular hypertrophy, torsade de pointes, and sudden cardiac death
Neuro-degenerative disease agents	Myocardial fibrosis (TGF-β) and adrenergic storm	Orthostatic hypotension, peripheral edema, and arrhythmias

## Data Availability

Not applicable.

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
