# Peer review of "Myocardial Cell Preservation from Potential Cardiotoxic Drugs: The Role of Nanotechnologies"

_pharmaceutics, 2022, doi:10.3390/pharmaceutics15010087_

Round 1

Reviewer 1 Report

Dear authors,

the review represents a brief and comprehensive overview on the use of nanotechnologies in the field of toxicology, particularly for prevention or reduction of cardiovascular side effects caused by cardiotoxic drugs. The review is well organized and of interest for the scientific audience. However, it should be improved in some aspects. Therefore, I would suggest minor revision, namely:

Point 1. The drugs in point 2 should be listed one after the other according to their pharmacological groups (for example cyclophosphamide whichi s an alkylating agent is listed between two antibodies) - is there any specific reason to list the drugs and groups in this order?

Point 2. In point 4 there is no discussion on how the problem could be solved - if there are no investigations and published results for these drugs, please note it but give also suggestions to overcome the problem by using nanotechnology based on the available state of the art.

Point 3. Point 5.2. can be expanded a little bit more as it is currently quite general. It is good to include more specific details about future innovative applications that would help solve the discussed problem.

Kind regards,

The reviewer

Reviewer 2 Report

The authors presented the paper "Myocardial cell preservation from potential cardiotoxic drugs: the role of nanotechnologies"

1) Much more fresh 2-3 years paper should be presented. The reference list may be improved. I understand that there are many good old works. However, it is difficult to show the progress in the area using the works older than 10 years.

2) The title of the review describes the role of nanotechnologies. However, in the abstract and the paper I see mostly liposomes, niosomes, dendrimers, lipid particles. Why you haven't discussed in the paper other important nanoparticles, nanocomposites? You can just google, and it will be numerous works. Moreover, if you want to limit the topic of the review, please limit the title and explain in the introduction section your meaning. In our way, insert new sections.

3) If you want to limit your review, please write the limitations in the Conclusion section. Moreover, the Conclusion section is too brief. Please, insert some outlook, perspectives, possible future investigation pathways, etc.

4) No any references are presented in the Introduction of section 1 showing the perspectives or problems of the area. I recommend more fully represent the issue of the review.

5) Finally, the review represents an essential topic. However, it is too brief. It doesn't contain enough pictures, tables summarizing the material.

Round 2

Reviewer 2 Report

Thank you for so hard work. I recommend only minor correction. Please, insert in section one magnetic nanoparticles (Fe3O4) (small subsection). It is an essential and popular topic. In this subsection, please use fresh references 2022 year. The most interesting thing is Fe3O4 bioinspared (coated with biomolecules, proteins) nanocomposites for biomedical applications. See the references, examples from MDPI search.

https://www.mdpi.com/search?sort=pubdate&page_count=50&q=magnetic+nanoparticles+biomedical&year_from=2022&year_to=2022&featured=&subjects=chem-materials%2Cmed-pharma%2Cbio-life&journals=magnetochemistry%2Cnanomaterials%2Cpharmaceutics%2Cmolecules%2Cencyclopedia&article_types=review-article%2CEntry&countries=
